# LOCAL PATCH AUTOAUGMENT WITH MULTI-AGENT COLLABORATION

## ABSTRACT

Data augmentation (DA) plays a critical role in improving the generalization of deep learning models. Recent works on automatically searching for DA policies from data have achieved great success. However, existing automated DA methods generally perform the search at the image level, which limits the exploration of diversity in local regions. In this paper, we propose a more fine-grained automated DA approach, dubbed Patch AutoAugment, to divide an image into a grid of patches and search for the joint optimal augmentation policies for the patches. We formulate it as a multi-agent reinforcement learning (MARL) problem, where each agent learns an augmentation policy for each patch based on its content together with the semantics of the whole image. The agents cooperate with each other to achieve the optimal augmentation effect of the entire image by sharing a team reward. We show the effectiveness of our method on multiple benchmark datasets of image classification, fine-grained image recognition and object detection (*e.g.*, CIFAR-10, CIFAR-100, ImageNet, CUB-200-2011, Stanford Cars, FGVC-Aircraft and Pascal VOC 2007). Extensive experiments demonstrate that our method outperforms the state-of-the-art DA methods while requiring fewer computational resources.

## 1 INTRODUCTION

Data Augmentation (DA) has been widely used to alleviate the overfitting risk in training deep neural networks by appropriately enriching the diversity of training data (Shorten & Khoshgoftaar, 2019; Gontijo-Lopes et al., 2020). Notable DA methods improve the performance and robustness of the neural networks, such as rotation, Mixup (Zhang et al., 2017) and Cutmix (Yun et al., 2019). But these approaches are typically handcraft and require human prior knowledge, which causes weak transferability of DA across different datasets. To relieve the dependence on manual design and further explore more adaptive augmentation, AutoAugment (AA) (Cubuk et al., 2018), as a new DA paradigm, is proposed to automate the search of the optimal DA policies (*i.e.*, DA operation, probability and magnitude) from the training dataset. To be specific, AA trains a proxy model with the augmentation policy generated by a controller, which is updated through reinforcement

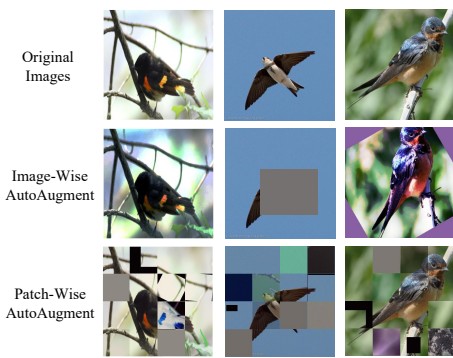

Figure 1: Illustration of different automated augmentation policies. We show the examples processed by image-wise automated DA, *e.g.*, AutoAugment (middle row) and processed by our proposed Patch AutoAugment (bottom row).

learning using validation accuracy as the reward signal. In spite of the superior performance of AA, its optimization procedure is computationally intensive due to the need to evaluate thousands of policies. Therefore, to reduce the search costs, multiple automated DA approaches (Lim et al., 2019; Ho et al., 2019; Hataya et al., 2019; Zhang et al., 2019b; Lin et al., 2019) have been proposed. For example, (Lim et al., 2019) employs density matching as a search method to accelerate the policy search, and (Zhang et al., 2019b) introduces adversarial learning to organize the target network training and augmentation policies search in an online manner.

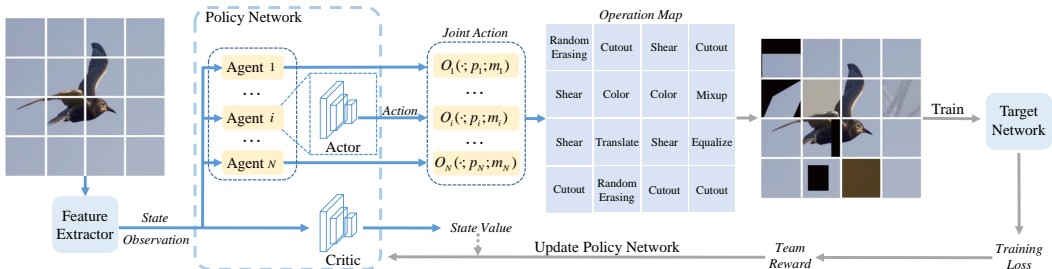

Figure 2: The framework of Patch AutoAugment (PAA). We divide an image into a grid of patches and assign an agent to each patch for selecting the optimal augmentation operation according to the patch content together with the whole image semantics. The agents cooperate with each other to achieve the joint optimal DA results by sharing a team reward through multi-agent reinforcement learning (MARL). Specifically, each agent outputs $O_i(\cdot; p_i; m_i)$ operation performed on patch $i$, which includes two parameters: probability of calling the operation $p_i$ and the magnitude $m_i$. Note that PAA is co-trained with the target network.

Yet the aforementioned automated DA methods all search for policies at the image level. They ignore the exploration of diversity in local regions, which may result in insufficient diversity of dataset and limit the benefits of DA (Gontijo-Lopes et al., 2020). In addition, due to this coarse-grained augmentation, they may damage key semantic information and introduce ambiguity into the training process (see a simple example in Figure 1 row 2, column 2). With those in mind, it is necessary to automatically search for the optimal augmentation policies for different regions by taking regional diversity into account. One straightforward idea is to directly apply image-wise automated DA methods in different regions. But such an intuitive solution ignores the contextual relationship between regions, which may lead to the non-globally optimal effectiveness of DA policies across the entire image. Besides, it may encounter an extremely high computational cost due to the need of optimizing multiple policies for regions respectively.

In this paper, we propose a new approach, named Patch AutoAugment (PAA) (see the last row of Figure 1), to address the above-mentioned problems. We first divide an image into a grid of patches to increase the flexibility of representations for different regions and take "patch" as the basic control unit. Then, we model the search for the augmentation policies of patches as a fully cooperative multi-agent task, and we leverage a multi-agent reinforcement learning (MARL) algorithm where agents cooperatively learn the policies. Specifically, based on the content of each patch and the semantics of the entire image, the agent searches for a policy in terms of choosing which transformation to apply out of the pre-defined DA operations. To encourage our policy networks to adaptively learn the beneficial augmentation policies for the target network, we use the feedback of the target network as the team reward signal to guide the policy networks to learn on the fly. All agents wind up benefiting from two mechanisms (*i.e.*, parameter sharing and centralized training with decentralized execution) in MARL. In this way, all agents collaboratively and parallelly learn policies to further achieve the joint optimal DA policy across the whole image and alleviate the computational cost.

In summary, our contributions can be summarized as follows: 1) We pinpoint that exploring diversity in local regions is important for automated learned DA approaches. To our best knowledge, we are the first to propose a more fine-grained automated DA approach, that searches for the optimal policies for the patches according to the content of the patch and the semantics of the entire image. 2) To further achieve the joint optimal policy across the image, we model the DA policy search of patches as a fully cooperative multi-agent task, and adopt a multi-agent reinforcement learning algorithm for Patch AutoAugment by considering the contextual relationship between the patches. 3) Our visualization results provide some insights to the DA community on which augmentation operation should be chosen for patches with different content during the whole training process.

## 2 RELATED WORK

### 2.1 DATA AUGMENTATION

Despite the remarkable performance of deep learning models in computer vision tasks, they often suffer from overfitting. Data augmentation (DA) as an effective technique has been proved to im-

prove the generalization ability of deep learning models (Shorten & Khoshgoftaar, 2019). Previous works (Cubuk et al., 2020; Gontijo-Lopes et al., 2020) indicate that the main benefit of DA arises from increasing the diversity of images. Popular techniques, such as rotation, flipping, color transformation, have been performed as commonly used augmentation methods. Recently, thanks to the skillful design of human experts, many DA methods (*e.g.*, Cutout (DeVries & Taylor, 2017), Mixup (Zhang et al., 2017), CutMix (Yun et al., 2019)) have been proposed and show significant performance. However, these manually designed methods require additional human prior knowledge on the dataset and sometimes they are limited to certain datasets and target tasks. Naturally, automatically finding DA methods from data have emerged to overcome the limitations of dependence on manual cumbersome exploration. Some works use generative adversarial networks (GANs) to directly generate training data (Shrivastava et al., 2017; Tran et al., 2017).

Furthermore, recent studies aim to automate the search for augmentation policies that choose the optimal transformations for training images. AutoAugment (AA) (Cubuk et al., 2018) adopts a controller to generate an augmentation policy which is used to train a proxy network, then gets the validation accuracy as the reward signal to update the controller using reinforcement learning. Unfortunately, the evaluation of thousands of policies makes AA computationally expensive. Therefore, multiple automated DA approaches focus on reducing the huge complexity and have achieved great progress. For example, PBA (Ho et al., 2019) employs hyperparameter optimization, Fast AA (Lim et al., 2019) uses a density matching algorithm, Adv AA (Zhang et al., 2019b) proposes an adversarial framework to jointly optimize the target network and the augmentation network, and DADA (Li et al., 2020) and DDAS (Liu et al., 2021a) target relaxing the discrete DA policy selection to a differentiable problem. Besides, RandAugment (Cubuk et al., 2020) removes the separate search on a proxy task in the offline automated DA methods (*e.g.*, AA) while also outperforming them. However, these automated DA methods perform the search at the image level, *i.e.*, they use the same policy on the whole image. It inevitably ignores the diversity of different regions in an image, which limits the diversity of data increased by data augmentation, and sometimes causes the damage of critical semantic information. In contrast, our method takes diversity in local regions and contextual relationships into account to search for augmentation policies for multiple regions.

Our proposed method is conceptually orthogonal to most region-based DA methods where DA transformations are performed in a non-automated way. For example, RandomErasing (Zhong et al., 2020), CutMix (Yun et al., 2019) perform cropping or replacement operation on a randomly selected rectangle region, Hard chips (Hong et al., 2019) extracts the object patches from an object pool and pastes them to an image to train the detector, and SECOND (Yan et al., 2018) performs DA by pasting the objects into the 3D point cloud. Some works further exploit class activation map (Zhou et al., 2016) or saliency map (Zhou et al., 2015) to select representative regions which are augmented by a randomly selected DA operation (*e.g.*, SaliencyMix (Uddin et al., 2020), SnapMix (Huang et al., 2020)), AugMix (Hendrycks et al., 2019), KeepAugment (Gong et al., 2020) and TrivialAugment (Müller & Hutter, 2021)). Yet our proposed method automatically searches for the augmentation transformations based on the given datasets and tasks.

## 2.2 MULTI-AGENT REINFORCEMENT LEARNING

The most significant characteristic of MARL is the cooperation between agents (Tampuu et al., 2017; Ma & Cameron, 2008; Foerster et al., 2017) which is distinct from directly applying reinforcement learning (RL) algorithms to multi-agent systems. Due to the limited observation and action of a single agent, cooperation is necessary in the reinforced multi-agent system to achieve the common goal. Compared with independent agents, cooperative agents can improve the efficiency and robustness of the model (Neto, 2005; Zhang et al., 2019a; Busoniu et al., 2008). Many vision tasks use MARL to interact with the public environment to make decisions, with the goal of maximizing the expected total return of all agents, such as image segmentation (Liao et al., 2020; Han et al., 2019; Ma et al., 2020), image processing (Furuta et al., 2019).

## 3 PROPOSED METHOD

As above mentioned, we aim to search augmentation policies for the patches to explore more augmentation diversity. Learning the optimal DA policies often needs to interact with the target network, which aligns with the nature of RL (*i.e.*, RL agents learn to make decisions through interacting with the environment (Sutton & Barto, 2018; Arulkumaran et al., 2017)). In addition, it is of great

necessity to take the patch content and the contextual relationship between patches into account. Therefore, we use multi-agent reinforcement learning (MARL) algorithm. Specifically, the search for patch policies is based on the content of the patch together with the semantics of the image, and policies are encouraged to coordinate to achieve the joint optimal DA policy across the whole image. In this section, we first describe the preliminaries of MARL, then elaborate on our augmentation policy formulation and modelling. Furthermore, we summarize the framework of Patch AutoAugment.

### 3.1 PRELIMINARIES OF MULTI-AGENT REINFORCEMENT LEARNING

We first introduce the preliminaries of reinforcement learning (RL). RL models the decision-making problem as a Markov decision process (MDP) which is presented with a tuple $(\mathcal{S}, \mathcal{A}, P, R, \gamma, T)$. In RL framework, given the state $s \in \mathcal{S}$, the agent takes an $a \in \mathcal{A}$ according to its policy $\pi(a\,|\,s)$ : $\mathcal{S} \times \mathcal{A} \to [0, 1]$ and then receives a reward $r : \mathcal{S} \times \mathcal{A} \to \mathbb{R}$. The environment moves to the next state with a transition function denoted as $P : \mathcal{S} \times \mathcal{A} \times \mathcal{S} \to [0, 1]$. $\gamma \in (0, 1]$ is a discount factor and $T$ is a time horizon. The agent aims to maximize the long-term reward $R$ over $T$ steps to learn the optimal policy $\pi^*$.

Furthermore, multi-agent reinforcement learning (MARL) considers a group of $N$ agents, denoted as $\mathcal{N}$, operating cooperatively in a shared environment towards a common goal. It can be formulated as a multi-agent MDP (MAMDP) (Boutilier, 1996) represented with a tuple $(\mathcal{S}, \{\mathcal{O}_i\}_{i=1}^N, \{\mathcal{A}_i\}_{i=1}^N, P, R, \gamma, T)$. Here, $\mathcal{S}$ describes the shared state space and $\{\mathcal{O}_1, \cdots, \mathcal{O}_N\}$ is a set of observations for agents. In the MARL configuration, each agent receives a private observation correlated with part of state, *i.e.*, $o_i : \mathcal{S} \to \mathcal{O}_i$. According to the global state $s$ and its observation $o_i$, the agent takes its action $a_i \in \mathcal{A}_i$ based on its policy $\pi_i(a_i|o_i, s)$, $\mathcal{A}_i$ is the action space for the $i$-th agent and $\mathcal{A} = \{\mathcal{A}_1, \cdots, \mathcal{A}_N\}$ denotes the joint action space. Then, the joint action $\boldsymbol{a} = a_1 \times \cdots \times a_N$ produces next state according to transition function $P : \mathcal{S} \times \mathcal{A} \times \mathcal{S} \to [0, 1]$ and the environment gives the team reward $r : \mathcal{S} \times \mathcal{A} \to \mathbb{R}$ to agents $\mathcal{N}$. The objective of each agent is to maximize the total accumulative reward $R$ to cooperatively learn the globally optimal policy $\boldsymbol{\pi^*} = \{\pi_1^*, \cdots, \pi_N^*\}$ that consists of the optimal policy of each agent. In this paper, as mentioned before, we employ MARL to search optimal policy for each patch, and achieve the optimal effectiveness of DA across the entire image to further improve the performance of the target network as possible.

### 3.2 PATCH AUTOAUGMENT

In our proposed Patch AutoAugment (PAA), we formulate the task of policy search for the patches as a cooperative multi-agent decision-making problem and adopt multi-agent reinforcement learning (MARL) to solve it. In the following, We clarify the detailed formulation $i$) the *state*, *observation* and *action* modeling for the policy, $ii$) an effective team reward function design and $iii$) the detailed MARL algorithm for policy learning) of Patch AutoAugment.

**Policy Modeling.** As illustrated in Figure 2, given the original input batch $\boldsymbol{x}$ and the corresponding label $\boldsymbol{y}$ (*i.e.*, $\{\boldsymbol{x}, \boldsymbol{y}\} = \{x_j, y_j\}_{j=1}^b$ and $b$ is the batch size), we divide an image $x_j$ into $N$ equal-sized and non-overlapping patches, denoted as $x_j = \{\mathcal{P}_j^i\}_{i=1}^N$ where $\mathcal{P}_j^i$ is the $i$-th patch of the image $x_j$. In our proposed method, we aim to search the augmentation policy for each patch. Therefore, in MARL formulation, the augmentation policy of the patch $\mathcal{P}^i$ is controlled by an agent $i \in \mathcal{N}$ and we detail the *state*, *observation* and *action* for the augmentation policy as below.

*State.* As aforementioned, the selection of augmentation operation for a patch is closely bound up with the contextual relationship between regions. Therefore, the augmentation policy needs to perceive the image semantics and we take the deep features of the whole image extracted by a backbone (*e.g.*, ImageNet pre-trained ResNet-18 (He et al., 2016)) as the global state $s$ which is visible to all agents. We analyze the impact of feature extractor initialization in Appendix F.

*Observation.* Apart from capturing state (*i.e.*, the global information), the agent only use their own observation (*i.e.*, the local information) which is invisible to other agents. The augmentation policy seeks to choose augmentation operation based on the content of the patch and the observation is generally part of the state. Considering all these factors, we utilize the deep features of the $i$-th patch as observation $o_i$, which are extracted by the same backbone as the state extractor.

*Action.* The augmentation policy is responsible for choosing which transformations to apply from pre-defined operations. Following the previous automated DA methods (Cubuk et al., 2018; Lim et al., 2019), we define the fifteen operation functions (*i.e.*, ShearX/Y, TranslateX/Y, Rotate, Invert, Equalize, Solarize, Posterize, Contrast, Color, Brightness, Sharpness, RandomErasing, Cutout, Mixup, Cutmix) to construct the action space $\mathcal{A}$. Given the state $s$ and the observation $o_i$, according to policy $\pi_i(a_i|o_i, s)$, each agent $i$ determines an action $a_i(\cdot) \in \mathcal{A}$, which is the operation performed on the patch. Each operation is associated with two hyperparameters: probability $p$ and magnitude $m$. In order to dramatically and effectively reduce the action space for augmentation policy, similar with (Cubuk et al., 2020; Ho et al., 2019; Lim et al., 2019), we take a fixed probability and magnitude schedule. Among them, the probability of applying the operation is sampled from the uniform distribution (*i.e.*, $p_i \sim U(0, 1)$. Following (Ho et al., 2019), we employ the same linear scale to be the magnitude schedule. In summary, the processed $i$-th patch in image $x_j$ is denoted as $\widetilde{P}_j^i = a_i(P_j^i)$ with the probability $p_i$ and the magnitude of $a_i(\cdot)$ is $m_i$, otherwise $\widetilde{P}_j^i = P_j^i$. Note that most operations are label-invariant, except Mixup (Zhang et al., 2017) and CutMix (Yun et al., 2019) are label-disturbing operations that combine different patches as well as their labels. We take Mixup as an example, then $\widetilde{\mathcal{P}}_j^i = \lambda \mathcal{P}_j^i + (1 - \lambda) \mathcal{P}_t^i$ with probability $p_i$, where $\mathcal{P}_t^i$ is a patch from another image $x_t, t \neq j$, $\lambda \sim Beta(\alpha, \alpha)$, for $\alpha \in (0, \infty)$, and the one-hot label is modified as $\widetilde{y}_j = \frac{\lambda}{N} y_j + \frac{(1-\lambda)}{N} y_t$. Once all patches are processed by the corresponding operations chosen by the augmentation policies, we obtain the image augmented by our PAA, denoted as $\widetilde{x}_j = \{\widetilde{\mathcal{P}}_j^1, \cdots, \widetilde{\mathcal{P}}_j^N\}$, and the final label $\widetilde{y}_j$. More operation details are shown in Appendix A.

**Reward Function.** The reward function is of importance to guide the agents to learn so that they follow desired behaviors. The previous work, AdvAA (Zhang et al., 2019b), attempts to increase the training loss of the target network to generate harder augmentation policies and explore the weakness of the target network. Inspired by AdvAA, we reformulate the reward design appropriately under our configuration. In our proposed PAA, the common objective of all agents is to improve the performance of the mainstream target task through enhancing the benefits of DA. Therefore, we compare the feedback of target network $\phi(\cdot)$ on the augmented data processed by our proposed PAA $\widetilde{x}$ with the original data $x$ and take their difference on the training losses as the reward for the policy in MARL, as in Eq. (1):

$$r = l(\phi(\widetilde{x}), \widetilde{y}) - l(\phi(x), y), \tag{1}$$

where $x$ and $y$ denote raw inputs and labels in supervision tasks. In our PAA model, all agents are encouraged to cooperate to achieve the common goal, thus we adopt the team reward function design (*i.e.*, the shared reward mechanism in MARL) as Eq. (1) for all agents to make the joint augmentation policy achieve the optimal effectiveness.

**Policy Learning.** Here, we introduce the training for the augmentation policies mentioned above. Considering that the action space is discrete, we adopt multi-agent Advantage Actor-Critic algorithm (Lowe et al., 2017) to learn the augmentation policies and encourage the coordination behaviors. In MARL, the framework of centralized training with decentralized execution (Foerster et al., 2018; Rashid et al., 2018) is generally adopted. More concretely, each agent $i$ has an actor which is to learn discrete policy $\pi_i(a_i|o_i, s)$ and agents share a common critic which aims to estimate the value of global state $V^{\boldsymbol{\pi}}(s)$. And we use the centralized critic to train decentralised actors. Here, we reformulate it appropriately for our task. We model the centralized action-value Q function that takes the actions of all agents in addition to state information $s$ and outputs the Q-value for the team, formulated as:

$$Q^{\boldsymbol{\pi}}(s, \boldsymbol{a}) = \mathbb{E}_\pi[R_t|s, a_1, \cdots, a_N], \tag{2}$$

where $\boldsymbol{a}$ is the joint action of all agents $\boldsymbol{a} = \{a_i, \cdots, a_N\}$ and $R_t = \sum_{l=0}^{T} \gamma^l r_{t+l}$ is the long-term discounted reward. Then, the advantage function on the augmentation policy is given as follows:

$$A^{\boldsymbol{\pi}}(s, \boldsymbol{a}) = Q^{\boldsymbol{\pi}}(s, \boldsymbol{a}) - V^{\boldsymbol{\pi}}(s). \tag{3}$$

The critic network is a Multi-layer Perceptron (MLP). And we use $\varphi$ to denote the its parameters. We take the square value of the advantage function $A^\pi$ as the loss function to update $\varphi$:

$$L(\varphi) = (A^{\boldsymbol{\pi}}(s, \boldsymbol{a}))^2. \tag{4}$$

Table 1: Test set accuracy (%) on CIFAR-10 and CIFAR-100. The results of our proposed PAA is the average accuracy ($\pm$standard deviation) over four random runs.

| Dataset | Model | Baseline | CutOut | Mixup | CutMix | Co-Mix | AA | FastAA | RA | DADA | PAA |
|---------|-------|----------|--------|-------|--------|--------|-----|--------|-----|------|-----|
| | WRN-28-10 | 96.1 | 96.9 | 97.1 | 97.2 | 97.3 | 97.4 | 97.3 | 97.3 | 97.3 | $\mathbf{97.5}_{\pm 0.1}$ |
| | SS(26 2x32d) | 96.4 | 97.0 | 97.2 | 97.3 | 97.4 | 97.5 | 97.3 | 97.5 | 97.3 | $\mathbf{97.6}_{\pm 0.1}$ |
| CIFAR-10 | SS(26 2x96d) | 97.1 | 97.4 | 97.7 | 97.8 | 98.0 | 97.7 | 97.7 | 97.8 | 98.0 | $\mathbf{98.1}_{\pm 0.1}$ |
| | SS(26 2x112d) | 97.2 | 97.4 | 98.0 | 98.0 | 98.0 | **98.1** | 98.0 | 98.0 | 98.0 | $\mathbf{98.1}_{\pm 0.1}$ |
| | Pyramid+SD | 97.3 | 97.7 | 98.0 | 98.1 | 98.2 | 98.5 | 98.2 | 98.5 | 98.3 | $\mathbf{98.6}_{\pm 0.1}$ |
| | WRN-28-10 | 81.2 | 81.6 | 82.1 | 82.8 | 83.4 | 82.9 | 82.7 | 83.3 | 82.5 | $\mathbf{83.4}_{\pm 0.3}$ |
| CIFAR-100 | SS(26 2x96d) | 82.9 | 84.0 | 85.4 | 85.6 | 85.8 | 85.7 | 85.1 | 85.5 | 85.7 | $\mathbf{85.9}_{\pm 0.2}$ |
| | Pyramid+SD | 86.0 | 87.8 | 88.5 | 88.9 | 89.0 | **89.3** | 88.1 | 89.2 | 88.8 | $89.2_{\pm 0.1}$ |

Table 2: Validation set Top-1 / Top-5 accuracy (%) on ImageNet.

| Method | Baseline | Mixup | CutMix | Co-Mix | AA | FastAA | RA | DADA | PAA |
|--------|----------|-------|--------|--------|-----|--------|-----|------|-----|
| ResNet-50 | 76.3/93.1 | 77.0/93.4 | 77.2/93.5 | 77.6/93.7 | 77.6/93.8 | 77.6/93.7 | 77.6/93.7 | 77.5/93.5 | $\mathbf{78.3}_{\pm 0.3}$ / $\mathbf{94.1}_{\pm 0.2}$ |
| ResNet-200 | 78.5/94.2 | 79.6/94.8 | 79.9/94.9 | 80.0/94.9 | 80.1/95.0 | 80.6/95.3 | 80.1/95.0 | 79.8/94.8 | $\mathbf{81.0}_{\pm 0.3}$ / $\mathbf{95.2}_{\pm 0.2}$ |

The global policy network is a fully convolutional network (FCN), denoted as $\theta$. Besides, to further achieve the ability to cooperate, similar to (Foerster et al., 2018; Rashid et al., 2018), the parameters of the actor networks of all agents are shared. The loss function for updating $\theta$ is defined as:

$$L(\theta) = -\log \pi_\theta(\boldsymbol{a}|s) A^{\boldsymbol{\pi}}(s, \boldsymbol{a}). \tag{5}$$

**Framework Summary.** In this part, we summarize the overall framework of our proposed PAA. As shown in Figure 2, PAA first divides an image into a grid of patches. Then, we use a feature extractor to obtain the deep features of the whole image as the state. Each agent draws its individual observation, *i.e.*, the deep features of the patch. According to the global state (*i.e.*, the semantics of the entire image) and the local observation (*i.e.*, the content of the patch), the actor networks output the augmentation operations of patches to further construct the joint operation map performed on the whole image. The augmented images are processed by our proposed PAA and then we input them to the target task network for parameters updating. Moreover, the feedback of the target network is used as the team reward signal to update the policy network.

## 4 EXPERIMENTS

### 4.1 EXPERIMENT OVERVIEW

In this section, to study the effectiveness of Patch AutoAugment (PAA), we conduct experiment on image classification, fine-grained image recognition and *local* tasks where local information is of importance, *e.g.*, object detection. Specifically, we focus on CIFAR-10, CIFAR-100 (Krizhevsky et al., 2009) and ImageNet (Deng et al., 2009) datasets as well as three fine-grained object recognition datasets, *i.e.*, CUB-200-2011 (Wah et al., 2011), Stanford Cars (Krause et al., 2013), FGVC-Aircraft (Maji et al., 2013) and Pascal VOC 2007 (Everingham & Winn, 2009). We describe the datasets in detail in Appendix B. We compare PAA with baseline pre-processing, Cutout (DeVries & Taylor, 2017), Mixup (Zhang et al., 2017), Cutmix (Yun et al., 2019), Co-Mixup (Co-Mix) (Kim et al., 2021), AutoAugment (AA) (Cubuk et al., 2018), Fast AutoAugment (FastAA) (Lim et al., 2019), RandAugment (RA) (Cubuk et al., 2020), DADA (Li et al., 2020) and Adversarial AA (AdvAA). The baseline follows (Zoph et al., 2018; Yamada et al., 2018; Gastaldi, 2017): standardizing the data, horizontally flipping with 0.5 probability, zero-padding and random cropping. Our policy model architecture design mainly follows (Chu et al., 2016), and we make some adjustments according to our task scenario. More details about our policy network architectures and the hyper-parameters of the target model and our policy model are supplied in Appendix C and D, respectively. Moreover, we default the number of patches $N$ to 16 except for CIFAR tasks $N = 4$. To ensure the reliability of our experiments, we run each experiment four times using different random seeds.

### 4.2 RESULTS AND ANALYSIS

**Classification Results on CIFAR-10 and CIFAR-100.** For CIFAR-10 and CIFAR-100, we examine on Wide-ResNet-28-10 (WRN-28-10) (Zagoruyko & Komodakis, 2016), Shake-Shake (SS) (Gastaldi, 2017) and Pyramid-Net+ShakeDrop (Pyramid+SD) (Han et al., 2017; Yamada et al., 2018) models. The results are reported in Table 1, which shows the proposed approach consistently outperforms several state-of-the-art DA methods. We observe that the improvement of performance is relatively slight, due to the small image size of CIFAR which is $32 \times 32$. In the following, we further apply our proposed PAA on datasets with larger image sizes and other networks.

Table 3: Comparison of computational cost (GPU hours) between our proposed PAA and other previous automated DA methods. We train Wide-ResNet-28-10 on CIFAR-10 and ResNet-50 on ImageNet. *Search*: the time of searching for augmentation policys. *Train*: the time of training the target network. The training time of PAA includes the inference time required to compute the state and observations. *Total*: the total time. Except PAA, all metrics are cited from (Zhang et al., 2019b; Lim et al., 2019).

| Dataset | GPU hours | AA | FastAA | AdvAA | PAA |
|---|---|---|---|---|---|
| | Search | 5000 | 3.5 | $\sim$**0** | $\sim$**0** |
| CIFAR-10 | Train | **6** | **6** | - | 7.5 |
| | Total | 5006 | 9.5 | - | **7.5** |
| | Search | 15000 | 450 | $\sim$**0** | $\sim$**0** |
| ImageNet | Train | **160** | **160** | 1280 | 270 |
| | Total | 15160 | 610 | 1280 | **270** |

Table 4: Test accuracy (%) on various fine-grained classification datasets including CUB-200-2011 (CUB) (Wah et al., 2011), Stanford Cars (Cars) (Krause et al., 2013) and FGVC-Aircraft (Aircraft) (Maji et al., 2013).

| Dataset | Model | Baseline | Mixup | CutMix | Co-Mix | AA | FastAA | RA | DADA | PAA |
|---|---|---|---|---|---|---|---|---|---|---|
| CUB | ResNet-50 | 85.5 | 86.2 | 86.1 | 87.1 | 86.8 | 86.5 | 86.9 | 86.8 | **87.5**$_{\pm 0.2}$ |
| | ResNet-101 | 85.6 | 87.7 | 87.9 | 88.2 | 88.1 | 87.9 | 88.0 | 88.1 | **88.3**$_{\pm 0.2}$ |
| Cars | ResNet-50 | 93.0 | 93.9 | 94.1 | 93.9 | 94.2 | 94.0 | 94.1 | 93.7 | **94.3**$_{\pm 0.1}$ |
| | ResNet-101 | 93.1 | 94.1 | 94.2 | 94.1 | 94.2 | 93.8 | 94.2 | 93.7 | **94.5**$_{\pm 0.1}$ |
| Aircraft | ResNet-50 | 91.0 | 92.0 | 92.2 | 92.2 | 92.3 | 92.2 | 92.3 | 91.8 | **92.6**$_{\pm 0.2}$ |
| | ResNet-101 | 91.6 | 92.9 | 92.3 | 93.1 | 92.8 | 92.9 | 92.6 | 92.8 | **93.5**$_{\pm 0.3}$ |

**Classification Results on ImageNet.** As shown in Table 2, we evaluate our method on ResNet-50 and ResNet-200 (He et al., 2016) backbone on ImageNet, and our PAA significantly improves the performance of the target networks. The results further demonstrate that our proposed method is an effective DA technique for consistent and expressive benefits for datasets with larger image sizes.

**Effectiveness of Fine-grained Classification.** Furthermore, we evaluate our proposed method on fine-grained image recognition tasks. According to previous work (Du et al., 2020; Chen et al., 2019), we take ResNet-50 and ResNet-101 as the backbones. The results are shown in Table 4, which illustrates that the performance of PAA is consistently better than other methods and PAA has achieved remarkable performance on these challenging fine-grained tasks.

**Effectiveness of Local Tasks.** In order to further verify the effectiveness of our proposed method, we conduct experiments on the object detection task. We adopt the mainstream detector Faster R-CNN (Ren et al., 2015) (ResNet-50) on the dataset Pascal VOC 2007 (Everingham & Winn, 2009) and show the mAP results on the Table 5. We observe that the proposed PAA achieves consistent improvements and PAA is indeed effective for the *local* tasks like object detection.

### 4.3 COMPLEXITY ANALYSIS

In this section, in order to further demonstrate the performance of PAA in terms of complexity, we compare the policy search time and training time of PAA with AA (Cubuk et al., 2018), FastAA (Lim et al., 2019) and AdvAA (Zhang et al., 2019b), as illustrated in Table 3. As shown in Table 3, compared to the previous works, PAA requires the fewest total computational resources, and the search time is almost negligible. As for the parameters, the total number of PAA model parameters (about 0.23M) is less than 1% of the target network (*e.g.*, ResNet50: about 25.5M).

In summary, the main methods to reduce the computational cost lie in three aspects: 1) The augmentation policy network is jointly optimized with the target network, similar to (Zhang et al., 2019b; Lin et al., 2019). Namely, our proposed method searches for policies in an online manner, obviating thousands of policies validation on a small proxy network and the requirement for retraining the target network. Besides, we use fixed schedules for the two corresponding hyperparameters (*i.e.*, probability and magnitude) of each transformation to effectively reduce the search space, which makes it easier to search for effective policies. By these means, PAA compresses most of **the policy search time** 2) As mentioned before, a MARL algorithm is adopted, in which all agents parallelly learn the augmentation policies, to reduce **the training time**. 3) Compared with the previous DA methods using image-by-image sequential transformations, we perform parallel transforms on tensor. Specifically, we pick the patches in a batch performing the same operation to reconstruct

Table 5: The mAP (%) results on Pascal VOC. We use the Faster R-CNN as the baseline detector.

| Method | Baseline | CutOut | Mixup | AA | PAA |
|--------|----------|--------|-------|------|------|
| mAP | 76.0 | 77.2 | 78.3 | 78.7 | **79.0** |

Table 6: Ablation study: performance comparisons (%) of *Patch RandomAugment* (PRA), *Patch Single AutoAugment* (PSAA) and our PAA on CIFAR-10 and CIFAR-100 (test accuracy) with Wide-ResNet-28-10 (WRN-28-10), and on ImageNet (Top-1 / Top-5 accuracy) with ResNet-50.

| Method | Policy | | | Dataset | | |
|--------|--------|------|------|---------|-----------|-----------|
| | random | SARL | MARL | CIFAR-10 | CIFAR-100 | ImageNet |
| PRA | ✓ | | | 97.1 | 83.0 | 77.9 / 93.9 |
| PSAA | | ✓ | | 97.3 | 83.1 | 78.0 / 93.9 |
| PAA(Ours) | | | ✓ | **97.5** | **83.4** | **78.3 / 94.1** |

a new tensor. And we use Kornia[1] to realize tensor transformations on GPU to further reduce **the processing time** which is included in the training time.

## 4.4 ABLATION STUDY

In this section, we study the effectiveness of MARL and also discuss the design of patch numbers, through several ablation experiments.

**Performance of random policies.** We randomize the augmentation policy for each patch, dubbed *Patch RandAugment* and compare it with our proposed PAA. To be specific, the augmentation policy is randomly selected from predefined transformations with uniform probability. As shown in Table 6, *Patch RandAugment* (PRA) leads to considerable appreciable improvements. However, our Patch AutoAugment with meaningful guidance significantly surpasses the random patch policies.

**Performance of the policies searched using single-agent RL.** Furthermore, we directly use the same single-agent RL to search for each patch's policy, termed as *Patch SingleAutoAugment* (PSAA) and compare it to our PAA, where the search task is a cooperative multi-agent problem. In short, PSAA ignores the contextual relationship between patches, where the agents are independent and patches are non-cooperative. The results (see Table 6) indicate that taking the contextual information into account with cooperative RL-agents has improved the joint effectiveness of DA.

**Impact of the number of patches.** We explore the effect of the number of patches on the target model performance. As shown in Table 7, we respectively set the number of patches $N = \{4, 16, 64, 256, 1024\}$. The results show that on ImageNet / CUB-200-2011, when patch numbers $N$ increase, the performance accuracy first increases and then decreases. In addition, when $N = 16$, PAA achieves the best performance. We analyze that the small number of patches may cause PAA to be unable to effectively explore local diversity, and its advantages would be limited. In particular, when $N = 1$, PAA degenerates into image-wise automated DA. In contrast, under too larger values (*e.g.*, the extreme case is to search for DA policy for each pixel), the local semantic consistency is broken and the benefit brought by the consideration of contextual relationship between patches is gradually overtaken.

## 4.5 VISUALIZATION

**Policy Visualization.** Grad-CAM (Selvaraju et al., 2017) is used to localize the important regions in the image. Therefore we can calculate the importance score of each patch, and we divide the importance scores into four bins, *i.e.*, *very important, important, normal* and *not important*. Then, we categorize patches into four groups and draw four stacked area charts showing the percentages of operations selected by PAA augmentation policies over time. We take ResNet-50 backbone trained on CUB-200-2011 as an example. Since our proposed PAA searches for patch policies in an online manner, the strategies change dynamically over time as shown in Figure 4. At the beginning of the training process, the selected actions are messy since the MARL network is in the exploratory stage. At the tail end of the training, the target network has converged, causing the percentage of all operations to be almost the same and the percentage to flatten out.

In addition, we have some interesting findings which may provide some insights to the DA community. *i*) The optimal augmentation strategies of patches vary by their importance levels. Therefore, it is necessary to take the image content into account when performing data augmentation. *ii*) In

---

[1]Kornia (Riba et al., 2020) is a differentiable computer vision library for PyTorch. We use it to accelerate augmentation operation on tensors.

Table 7: Ablation study: performance comparisons (Top-1 accuracy (%)) under five different patch numbers $N$ on CIFAR-10 (image size: $32 \times 32$) with WRN-28-10 and on ImageNet (image size: $224 \times 224$) together with CUB-200-2011 (CUB) (image size: $448 \times 448$) with ResNet-50 backbone.

| N | 4 | 16 | 64 | 256 | 1024 |
|---|---|----|----|-----|------|
| CIFAR-10 | **97.5** | 97.3 | 97.2 | 97.1 | - |
| CUB | 87.2 | **87.5** | 87.3 | 87.1 | 86.8 |

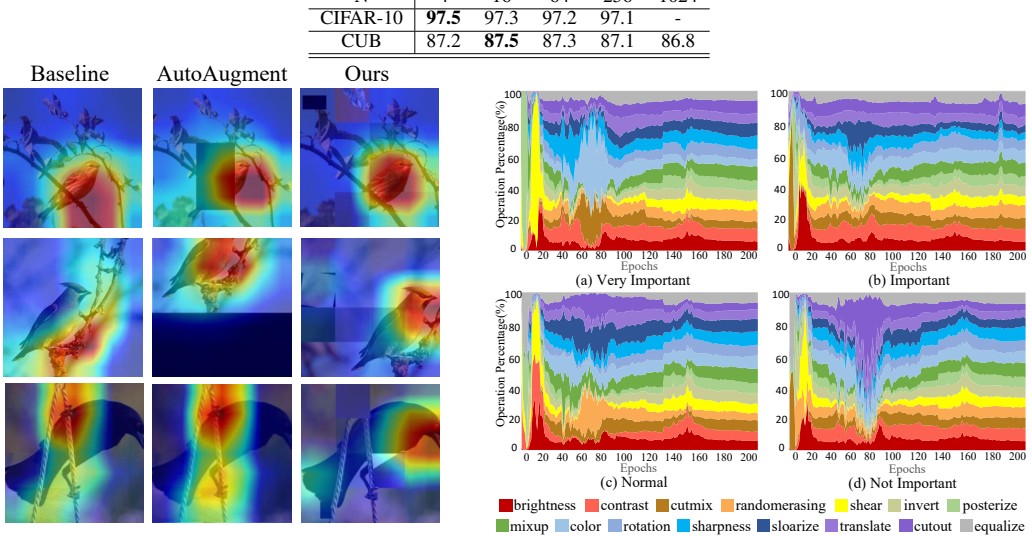

Figure 3: Grad-CAM visualization on the examples from CUB with ResNet-50 trained by baseline, AutoAugment and our PAA. PAA tends to help the target network focus on task-related features.

Figure 4: We categorize patches into four groups according to the patch importance calculated through Grad-CAM. We draw four **stacked area chart of the operations' percentages** over time. The area represents the percentage of each operation.

the middle of the training process, different types of patches prefer to select different augmentation operations. Concretely, as illustrated in Figure 4, for the important patches, *color* transformation is mostly picked. For the unimportant patches, *RandomErasing* and *Cutout* are usually chosen by PAA. Important patches commonly take along semantic information that is related to mainstream tasks. It is better to choose the mild transformations (*e.g.*, *color*) for them, which can effectively protect the semantic information from being damaged. In contrast, unimportant patches typically carry unexpected features (Singla et al., 2021) which are causally unrelated to the desired class. Severe transformations (*e.g.*, *Cutout*, RandomErasing) could be chosen for them, which introduce noise and disturbance to reduce the impact of unexpected features on the target network learning.

**Grad-CAM Visualization.** Here, we adopt Grad-CAM (Selvaraju et al., 2017) to visualize the learned features to intuitively show the impact of PAA, as shown in Figure 3. We take the ResNet-50 as the backbone which is trained with dataset processed by 1) Baseline 2) AutoAugment and 3) Patch AutoAugment, respectively. We observe that the model trained with PAA focus on more task-related areas rather than spurious correlations (*e.g.*, the branch where the bird stands) or overemphasized features (*e.g.*, birds' claws). More visualization results are provided in the Appendix E.

## 5 CONCLUSION

In this paper, we propose Patch AutoAugment (PAA), a more fine-grained automated data augmentation approach. Our method adopts multi-agent reinforcement learning to automatically search for the optimal augmentation policies for patches, and encourages agents to cooperate with each other to further achieve the joint optimal policy across the entire image. Extensive experiments demonstrate that PAA improves the target network performance with low computational cost in various tasks. Meanwhile, we use visualization to show that PAA is beneficial for the target network to localize more class-related cues. Furthermore, we hope that our visual observations of policies will be useful for future development. In future work, we will investigate different schemes on dividing different regions. Additionally, our method is naturally aligned with the patch token mechanism in the current vision transformers (Dosovitskiy et al., 2020; Touvron et al., 2021; Liu et al., 2021b) and the data augmentation specific to vision transformers has not been extensively studied. Therefore, we leave the automated data augmentation for vision transformers to future work, which may provide some interesting insights to the community.

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

Table 8: We list fifteen kinds of augmentation operations that we use. Additionally, the schedule of magnitude (*i.e.*, the parameters for Kornia (Riba et al., 2020) pytorch library) for each operation are shown in the third column. Some transformations do not use the magnitude information (*e.g.*, CutMix and Cutout).

| Operation Name | Description | magnitudes |
|---|---|---|
| Brightness | Adjust the brightness of the patch. A magnitude=0 gives a black patch, whereas magnitude=1 gives the original patch. | brightness=(0.5, 0.95) |
| Contrast | Control the contrast of the patch. A magnitude=0 gives a gray patch, whereas magnitude=1 gives the original patch. | contrast=(0.5, 0.95) |
| CutMix | Replace this patch with another patch (selected at random from the patches which are also performed *CutMix*). | - |
| Cutout | Set all pixels in this patch to the average value of the patch. | - |
| Invert | Invert the pixels of the patch | - |
| Mixup | Linearly add the image with another image (selected at random from the patches which are also performed *Mixup*). | $\lambda \sim Beta(1, 1)$ |
| Posterize | Reduce the number of bits for each pixel to magnitude bits. | bits=3 |
| Solarize | Invert all pixels above a threshold value of magnitude. | thresholds=0.1 |
| RandomErasing | Erases a random rectangle region in a patch. | scale=(0.09, 0.36), ratio=(0.5, 1/0.5) |
| Rotation | Rotate the patch magnitude degrees. | degrees=30.0 |
| Sharpness | Adjust the sharpness of the image. A magnitude=0 gives a blurred image, whereas magnitude=1 gives the original image. | sharpness=0.5 |
| Shear(X/Y) | Shear the image along the horizontal or vertical axis with rate magnitude | shear=(-30, 30) |
| Translate(X/Y) | Translate the patch in the horizontal or vertical direction by absolute fraction of patch length. | translate=(0.4, 0.4) |
| Color | Adjust the color balance of the image. | hue=(-0.3, 0.3) |
| Equalize | Equalize the image histogram. | - |

Table 9: The model architecture of policy network and critic network in PAA. For each convolution layer, we list the input dimension, output dimension, kernel size, stride, and padding. For the fully-connected layer, we provide the input and output dimension. BN is short for batch normalization.

| Layer | Policy network | Critic network |
|---|---|---|
| 1 | ReLU,Conv2D(32,64,3,1,1),BN | FC(1568,256) |
| 2 | ReLU,Conv2D(64,64,3,1,1),BN | ReLU |
| 3 | ReLU,Conv2D(64,15,3,2,1),Softmax | FC(256,1) |

## APPENDIX

## A   OPERATIONS DETAILS

Following (Cubuk et al., 2018; Ho et al., 2019; Lim et al., 2019; Zhang et al., 2019b), we define the fifteen common augmentation operations to form the action space. Here, we detail the description of these operations, as illustrated in Table 8. In addition, we give magnitudes range of augmentation operations corresponding to hyperparameters of the functions in the Kornia PyTorch library. Some operations, such as Cutout and CutMix, have no parameters.

Additionally, in our implementation, we pick out the patches that perform the same operation and put them into a new tensor. Then, we speed up the process by performing parallel transformations on the tensor. Tensor transformations on GPU can be realized by Kornia (Riba et al., 2020) to reduce the computational costs. In particular, when performing the label-disturbing operations (*i.e.*, Mixup and CutMix), a patch needs to mix with another patch that is randomly selected from the new tensor. Namely, a patch is mixed with another patch that performs the same operation.

## B   DATASETS

We evaluate Patch AutoAugment (PAA) on the following datasets: CIFAR-10 (Krizhevsky et al., 2009), CIFAR-100 (Krizhevsky et al., 2009), ImageNet (Deng et al., 2009) and three fine-grained

Table 10: The hyperparameters of various target models on CIFAR-10, CIFAR-100, ImageNet, CUB-200-2011, Stanford Cars and FGVC-Aircraft. LR represents learning rate of the target network, WD represents weight decay, and LD represents learning rate decay method. If LD is multistep, we decay the learning rate by 10-fold at epochs 30, 60, 90 etc. according to LR-step. LR-A2C represents the learning rate of augmentation model.

| Dataset | Model | BatchSize | LR | WD | LD | LRstep | LR-A2C | Epoch |
|---------|-------|-----------|-----|-----|-----|--------|--------|-------|
| CIFAR-10 | WRN-28-10 | 128 | 0.1 | 5e-4 | cosine | - | 1e-3 | 200 |
| | SS(26 2x32d) | 128 | 0.2 | 1e-4 | cosine | - | 1e-4 | 600 |
| | SS(26 2x96d) | 128 | 0.2 | 1e-4 | cosine | - | 1e-4 | 600 |
| | SS(26 2x112d) | 128 | 0.2 | 1e-4 | cosine | - | 1e-4 | 600 |
| | Pyramid+SD | 128 | 0.1 | 1e-4 | cosine | - | 1e-4 | 600 |
| CIFAR-100 | WRN-28-10 | 128 | 0.1 | 5e-4 | cosine | - | 1e-4 | 200 |
| | SS(26 2x96d) | 128 | 0.1 | 5e-4 | cosine | - | 1e-4 | 1200 |
| | Pyramid+SD | 128 | 0.5 | 1e-4 | cosine | - | 1e-4 | 1200 |
| ImageNet | ResNet-50 | 512 | 0.1 | 1e-4 | multistep | [30,60,90,120,150] | 1e-4 | 270 |
| | ResNet-200 | 512 | 0.1 | 1e-4 | multistep | [30,60,90,120,150] | 1e-4 | 270 |
| CUB-200-2011 | ResNet-50 | 512 | 1e-3 | 1e-4 | multistep | [30,60,90] | 1e-4 | 200 |
| | ResNet-101 | 512 | 1e-3 | 1e-4 | multistep | [30,60,90] | 1e-4 | 200 |
| Stanford Cars | ResNet-50 | 512 | 1e-3 | 1e-4 | multistep | [30,60,90] | 1e-4 | 200 |
| | ResNet-101 | 512 | 1e-3 | 1e-4 | multistep | [30,60,90] | 1e-4 | 200 |
| FGVC-Aircraft | ResNet-50 | 512 | 1e-3 | 1e-4 | multistep | [30,60,90] | 1e-4 | 200 |
| | ResNet-101 | 512 | 1e-3 | 1e-4 | multistep | [30,60,90] | 1e-4 | 200 |

image recognition datasets (CUB-200-2011 (Wah et al., 2011), Stanford Cars (Krause et al., 2013), FGVC-Aircraft (Maji et al., 2013)) and Pascal VOC (Everingham & Winn, 2009).

To be specific, both CIFAR-10 and CIFAR-100 have 50,000 training examples. Each image of size $32 \times 32$ belongs to one of 10 categories. ImageNet dataset has about 1.2 million training images and 50,000 validation images with 1000 classes. Original ImageNet data have different sizes and we resize them to $224 \times 224$. In addition, we evaluate the performance of our proposed PAA on three standard fine-grained object recognition datasets. CUB-200-2011 (Wah et al., 2011) consists of 6,000 train and 5,800 test bird images distributed in 200 categories. Stanford Cars (Krause et al., 2013) contains 16,185 images in 196 classes. The FGVC-Aircraft dataset contains 10,200 images of aircraft, with 100 images for each of 102 different aircraft model variants, most of which are airplanes. The image size in the above three datasets is $448 \times 448$. The Pascal VOC dataset consists of a set of images and each image has an annotation file giving a bounding box and object class label for each object in one of the twenty classes present in the image. The data has been split into 50% for training and 50% for testing. In total there are 9,963 images, containing 24,640 annotated objects.

## C  MODEL ARCHITECTURE

Here, we provide the detailed model architecture for each component in our PAA augmentation policy model, including feature extractor network, actor network and critic network. We use the pretrained on ImageNet ResNet-18 backbone (excluding the final avgpool and softmax layer) to extract the deep features of the image and the patch, which are denoted as the state $s$ and the observation $o_i$ respectively. As for the global policy network and the critic network, the detailed model architectures are shown in Table 9. The policy network is a fully convolutional neural network (FCN). The parameters of the actor networks of all agents are shared. Although they share parameters, each agent's policy is not identical. When determining an action, each agent's policy relies on different activations of the global policy network.

## D  HYPERPARAMETERS

We detail various target models hyperparameters (*e.g.*, batch size, learning rate and training epochs) on CIFAR-10, CIFAR-100, ImageNet, CUB-200-2011, Stanford Cars and FGVC-Aircraft in Table 10. We do not specifically tune these hyperparameters, and all of these are consistent with previous works (Cubuk et al., 2018; Lim et al., 2019; Dabouei et al., 2021; Du et al., 2020; Chen et al., 2019).

Table 11: Ablation study: impact of the initialization of the feature extractor. We respectively use the pretrained ResNet-18 and non-pretrained ResNet-18 as the feature extractor to get the deep features as the state/observation and show the test accuracy (%).

| Dataset & Model | Baseline | Pretrained ResNet-18 | Non-pretrained ResNet-18 |
|---|---|---|---|
| CIFAR-100 (WRN-28-10) | 81.2 | 83.4 | 83.2 |

In addition, the time horizon $T$ determines the times of augmentations performed sequentially on a patch. Therefore the agent can output more than one augmentation for a patch by setting $T > 1$. In our PAA model, we set time horizon $T = 1$, *i.e.*, the augmentation policies take actions at every time step for saving computational cost and avoiding over-regularization. Even in the single-step decision-making process (*i.e.*, $T = 1$), RL can still effectively search for the optimal policy as illustrated in (Mnih et al., 2016; Singh et al., 2000). In addition, we use the SGD optimizer with an initial learning rate of 1e-4 to train the actor network and the critic network.

## E  MORE GRAD-CAM VISUALIZATION

In this section, we provide more Grad-CAM (Selvaraju et al., 2017) results of ResNet-50 models trained using baseline augmented data, AA (Cubuk et al., 2018) and our proposed PAA, respectively, as shown in Figure 5. The visualization results demonstrate that our proposed PAA improves the localization ability of the target network and tends to help the target network focus on more parts of the foreground object. In short, our proposed PAA make the target network focus on the important or representative regions closely related to the class within an image.

## F  IMPACT OF THE INITIALIZATION OF THE FEATURE EXTRACTOR

We further study the impact of the initialization of the feature extractor as shown in Table 11. We first observe that the performance of the *Non-pretrained ResNet-18* scheme (using *non-pretrained ResNet-18* as the state/observation extractor) is still much better than the baseline. In other words, our agents do not rely on the pre-trained ResNet to produce features to be used as the state or observation. We also notice that the performance of the *Non-pretrained ResNet-18* scheme is slightly lower than the *Pretrained ResNet-18*. This may be because that at the very early stage of the training, the features extracted by *Non-pretrained ResNet-18* are not as good as *Pretrained ResNet-18* in practice.

## G  DETAILS OF COMPUTATIONAL COST

The computational cost of our proposed PAA is estimated on GeForce GTX 1080 Ti while AA (Cubuk et al., 2018) and AdvAA (Zhang et al., 2019b) are on NVIDIA Tesla P100. Besides, FastAA (Lim et al., 2019) is estimated on NVIDIA Tesla V100. The computational cost is mainly used for searching policies and training the target networks.

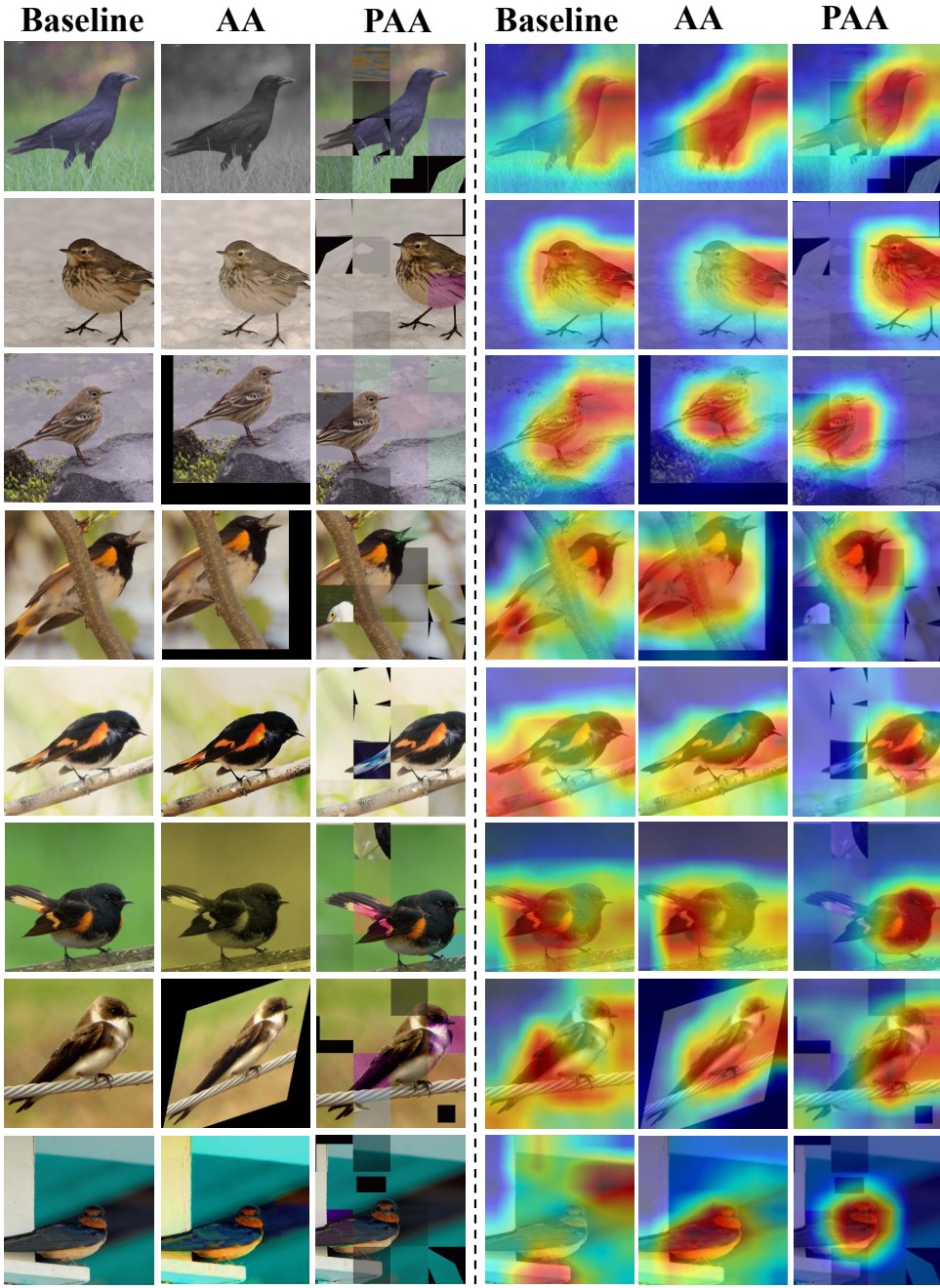

Figure 5: We give more visualizations of augmentation and Grad-CAM(Selvaraju et al., 2017) results of baseline, AA(Cubuk et al., 2018) and our PAA. PAA performs rich diversity of DA and tends to help the target model focus on the class-related features.

