# OpenReview forum: "Local Patch AutoAugment with Multi-Agent Collaboration"
_ICLR.cc/2022/Conference — ICLR 2022 Submitted_

### Official Review · Reviewer_cJ8q · 2021-11-01

**Correctness:** 3
**Technical Novelty And Significance:** 2
**Empirical Novelty And Significance:** 3
**Recommendation:** 6
**Confidence:** 4

**Main Review:**

### Strengths:
- 1.	The paper conducted a comprehensive evaluation of their approach over several datasets. They achieve gains over the compared baseline method validating the efficacy of their proposed approach.
- 2.	The approach considers patched based augmentation, which is sufficiently different/novel from existing work.

### Weaknesses:
- 3.	The motivation of the approach is somewhat lacking. For example, it is unclear why MARL on patches is much faster. I think the paper could benefit from motivating more on why MARL and what is its benefit with respect to the task.
- 4.	The paper did not reference or compared to a key related work, “Differentiable automatic data augmentation” [A]. Based on its relevance, the paper should discuss and contrast with this work, as they also have minimal search time. Specifically, comparison in Table 2 and Table 3 would strengthen the paper.
- 5.	How is the model architecture of the actor/critic network determined? Also, what ranges of hyperparameters for MADDPG are considered? I saw the hyperparameters for the target model but not MADDPG. RL is known to be sensitive to hyperparameters, I wonder if the authors will release the code and pre-trained models for reproducibility.
- 6.	Ablation with independent agents as a baseline would be interesting. It would further motivate the use of MADDPG and the multi-agent setting.

### References
[A] Li, Yonggang, et al. "Differentiable automatic data augmentation.", Proc. ECCV, 2020.




**Summary Of The Paper:**

This paper proposes an automatic data augmentation approach. Different from existing works, they proposed to augment patches in the image rather than the whole image. The approach is formulated as a multi-agent reinforcement learning problem. They empirically show the effectiveness of their approach across several image classification datasets.

**Summary Of The Review:**

The main concern with the paper is with the motivation and missing reference/comparison with a key existing work.
Currently, I recommend a weak reject.

---

> ### Author Response · Authors · 2021-11-15
> **Response to Reviewer cJ8q**
>
> Thank you for your helpful feedback. We would like to address your concerns below.
>
> - **Q1:** Why use MARL and what are its benefits for the task and why the use of MARL is much faster?
>
>     **A1:** - **Why use MARL  and what are its benefits for the task?**
>   Our method aims to search for the optimal augmentation policies for patches.  Since the patches in an image are not isolated, it is necessary to take the patch content and the contextual relationship between patches into account. We leverage the MARL algorithm where the agents cooperatively learn the augmentation policies to achieve the global optimal effectiveness of DA policies across the entire image. In addition, we compare the performance between MARL and SARL (*i.e.,* the independent agents respectively learn DA policies for patches without cooperation), as shown in Table 5, to experimentally show the benefits of MARL.
>
>     **- Why the use of MARL is much faster?**
>   The use of independent agents to learn the policies for patches means that it needs to optimize multiple policies for regions respectively. On the contrary, we adopt the MARL algorithm to reduce the training time, where all agents parallelly and simultaneously learn the augmentation policies.  Therefore, the MARL algorithm encounters lower computational costs than SARL.
>
> &nbsp;
> - **Q2: Compare and discuss with DADA [1].**
>
>     **A2:** Thank you for the suggestions. DADA [1] aims to reduce computational cost in AA by relaxing the discrete DA policy selection to a differentiable problem. While our method PAA focuses on exploring the diversity of local regions to search DA policies for patches in an online manner. We further compare the performance between DADA [1] and our PAA in the table below. And we have discussed DADA [1] in more detail and included the comparison results in the revision.
>
>     |  Dataset  |    Model     | DADA |   PAA    |
>     | :-------: | :----------: | :--: | :------: |
>     | CIFAR-10  |  WRN-28-10   | 97.3 | **97.5** |
>     | CIFAR-10  | SS(26 2x96d) | 98.0 | **98.1** |
>     | CIFAR-100 |  WRN-28-10   | 82.5 | **83.4** |
>     | CIFAR-100 | SS(26 2x96d) | 85.7 | **85.9** |
>     | ImageNet  |  ResNet-50   | 77.5 | **78.3** |
>     |    CUB    |  ResNet-50   | 86.8 | **87.5** |
>     |    CUB    |  ResNet-101  | 88.1 | **88.3** |
>     | Aircraft  |  ResNet-101  | 92.8 | **93.5** |
>
> &nbsp;
> - **Q3: "How is the model architecture of the actor/critic network determined? Also, what ranges of hyperparameters for MADDPG are considered? I saw the hyperparameters for the target model but not MADDPG. RL is known to be sensitive to hyperparameters, I wonder if the authors will release the code and pre-trained models for reproducibility."**
>
>     **A3:** Our model architecture design mainly follows [2, 3], and some adjustments are made according to the task scenario. As for the hyperparameters for the policy network, we use the SGD optimizer with a learning rate of 1e-4 to train the actor and the critic network. The details of the model architecture and the hyperparameters of our policy network are provided in Appendix C & D. Once the ICLR review process is completed, we will release the code and pre-trained models for reproducibility.
>
> &nbsp;
> - **Q4: Ablation with independent agents as a baseline would be interesting.**
>
>     **A4:** We have shown the performance comparisons of independent agents in Table 5. Patch Single AutoAugment (PSAA) searches the policies for patches using single-agent RL where the agents are independent. The results in Table 5 show that independent RL agents are not better than cooperative RL agents, and also reflex the advantages of using MARL.
>
> &nbsp;
>
> **Reference**
>
> [1] Li Y, Hu G, Wang Y, et al. DADA: Differentiable automatic data augmentation[J]. arXiv preprint arXiv:2003.03780, 2020.
>
> [2] Chu T, Qu S, Wang J. Large-scale multi-agent reinforcement learning using image-based state representation[C]//2016 IEEE 55th Conference on Decision and Control (CDC). IEEE, 2016: 7592-7597.
>
> [3] Gupta S, Hazra R, Dukkipati A. Networked multi-agent reinforcement learning with emergent communication[J]. arXiv preprint arXiv:2004.02780, 2020.

---

> > ### Comment · Reviewer_cJ8q · 2021-11-21
> > **Reviewer Response**
> >
> > Thanks for addressing my comments and updating the paper.
> > I have updated my recommendation.

---

> > > ### Author Response · Authors · 2021-11-21
> > > **Thank you**
> > >
> > > Thank you for your positive feedback! We appreciate your kind support!

---

### Official Review · Reviewer_XdSw · 2021-11-01

**Correctness:** 4
**Technical Novelty And Significance:** 3
**Empirical Novelty And Significance:** 2
**Recommendation:** 6
**Confidence:** 3

**Main Review:**

## Strengths

  - S1: Ablation studies and theoretical explanations
    motivate the choice for multi-agent reinforcement
    learning reasonably well.
  - S2: The efficiency of the proposed method is
    investigated very clearly in Section 4.3. It seems
    considerably more efficient than many other
    learning-based DA methods.
  - S3: The patch-wise data augmentation is a powerful idea
    which can spark a large body of interesting follow-up
    research.


## Limitations

  - L1: While this is not the main focus of the paper, the
    empirical improvement over RL baselines like Fast
    AutoAugment and non-RL baselines like RandAugment is
    relatively small.
  - L2: (Minor) While multiple datasets are used for
    evaluation, only two Resnet variants are investigated
    for ImageNet. It would be nice to see an evaluation of
    the proposed data augmentation procedure on other NN
    architectures as well on ImageNet, but I also
    understand that the computational costs of this can be
    quite large, so it may not always be possible.
  - L3: (Minor) The agents rely on a pre-trained ResNet to
    produce features to be used as observations.


## Suggestions (Per-Section)

### Related Work

  - This is minor, but it may be worth briefly mentioning
    [hong2019patch]. While their work doesn't do patch
    level data augmentation, and instead "patches" objects
    from an existing bank onto training samples, it still
    has connections to things like CutMix that make it
    relevant.
  - Some recent papers on automatic augmentation are
    missing and would be worth discussing. This includes
    [DADA], [DDAS], [hendrycks2019augmix] and
    [mueller2021trivialAugment]. In particular, Mueller and
    Hutter's paper highlights an interesting limitation of
    their otherwise robust baseline, which is that it
    doesn't work too well for object detection, an area
    where the current method may have the upper hand!
  - Similarly, there are some further papers, such as […]
    that perform a kind of "local" (though not strictly
    patch-based) data augmentation by "pasting in" objects
    into the 3D point cloud.


### Method

  - "directly applying reinforcement learning (RL)
    algorithm to …" => "algorithms"
  - "In MARL configuration" => "In the MARL configuration"
  - Towards the end of Section 3.2, it may make more sense
    to replace the "Until all patches" to "Once all
    patches…" or maybe "After".
  - Q1: Can an agent output more than one augmentation for
    a patch? Would it be possible for a patch to, e.g., get
    rotated AND colorized?
  - Q2: Given the more "local" nature of this augmentation
    method, have you considered evaluating it on more
    "local" tasks, such as object detection and semantic
    segmentation. Perhaps it could make a bigger impact
    there.
  - Q3: Given that the parameters of actor networks of all
    agents are shared (see above Eq. (5)), does that mean
    that each actor's policy is identical, and the only
    difference between the actors is what observation each
    one gets (the local patch features)?


### Experiments

  - At the start of Section 4.1, "Exactly, we focus" =>
    "Specifically, we focus"
  - In "Policy Visualization", "important levels" ->
    "importance levels"
  - In Section 4.3 I would say the "main methods for
    reducing the computational cost", since what you next
    describe are methods for reducing the cost, not what
    one would typically call reasons.
  - Table 3 is a little hard to follow because of the
    different hardware employed by each column. After
    staring at it for a while it's becoming a bit clearer,
    but at first glance it was a bit confusing.
  - Q4: Do the times from Table 3 include the inference
    time required to compute the agent state and
    observations with the auxiliary ResNet-18?


### Conclusion

  - The parallels to vision transformers are very
    interesting and help give the reader a broader
    perspective on possible applications of the current
    paper.
  - (Minor) The word "Furthermore" is repeated for two
    almost consecutive sentences. Using something like
    "additionally" to start one of the sentences can
    improve the flow of the paragraph a little.


## References

  - [hong2019patch]: Hong, Sungeun, Sungil Kang, and
    Donghyeon Cho. "Patch-level
  augmentation for object detection in aerial images."
  Proceedings of the IEEE/CVF International Conference on
  Computer Vision Workshops. 2019.
  - [cubuk2020randaugment]: Cubuk, Ekin D., et al.
    "Randaugment: Practical automated
  data augmentation with a reduced search space."
  Proceedings of the IEEE/CVF Conference on Computer Vision
  and Pattern Recognition Workshops. 2020.
  - [DDAS] Liu, Aoming, et al. "Direct Differentiable
    Augmentation Search." arXiv preprint arXiv:2104.04282
    (2021).
  - [DADA] Yonggang Li, Guosheng Hu, Yongtao Wang, Timothy
    M. Hospedales, Neil Martin Robertson, and Yongxing
    Yang. DADA: differentiable automatic data augmentation.
    CoRR, abs/2003.03780, 2020.
  - [mueller2021trivialAugment]: Müller, Samuel G., and
    Frank Hutter. "TrivialAugment: Tuning-free Yet
    State-of-the-Art Data Augmentation." ICCV 2021.
  - [hendrycks2019augmix]: Hendrycks, Dan, et al. "Augmix:
    A simple data processing method to improve robustness
    and uncertainty." arXiv preprint arXiv:1912.02781
    (2019).


**Summary Of The Paper:**

  The paper proposes an evolution of the traditional
  pipeline of image data augmentation used to reduce ML
  model overfitting.

  Instead of applying transformations such as shear,
  rotate, CutOut, etc. at the image level, the proposed
  technique divides the images into a fixed grid and
  applies a potentially different transformation to each
  cell. The problem of selecting a transform for each cell
  is cast as a multi-agent RL (MARL) task, and the agents
  learn as the main network trains within a (multi-agent)
  Advantage Actor Critic framework.

  The agents use a shared reward mechanism, with the reward
  defined as the difference between the loss on the
  augmented sample and the original loss.

  The regular grid is fixed for a dataset and
  hyperparameters like the magnitude of the augmentations
  follow a fixed schedule; the agents only pick _which_
  augmentation to apply on a patch.

  Experiments performed on CIFAR-{10,100}, ImageNet,
  CUB-200-2011, Stanford Cars, and FGVC-Aircraft show
  relatively small but very consistent improvements in
  terms of image classification accuracy. Different design
  choices (MARL vs. single-agent vs. random, grid size,
  etc.) are ablated and discussed in detail.


**Summary Of The Review:**

  The paper proposes a new way of looking at the classic
  task of image augmentation for classification. Instead of
  learning an image-level augmentation policy using RL, the
  proposed approach uses multi-agent RL to learn
  patch-specific augmentations. While the benefit of
  multi-agent RL is relatively small, it seems significant
  and the general idea of learning to apply different
  augmentations to different parts of the image is
  interesting and, to the best of my knowledge, new. The
  idea to apply image augmentations at a local level is the
  core contribution of this paper, in my opinion.

  While it would be interesting to see how this technique
  works on tasks which, unlike image-level classification,
  are themselves more "local", such as semantic
  segmentation or detection, overall the proposed approach
  is evaluated thoroughly.

  The writing is clear and while the authors don't
  explicitly mention an intent to publish their source
  code, it should be possible to implement the proposed
  approach just based on the description from the paper. As
  such, I recommend accepting the paper to ICLR.

---

> ### Author Response · Authors · 2021-11-15
> **Response to Reviewer XdSw**
>
> Thank you for the constructive comments and positive feedback! We will incorporate your suggestions in the revision.
>
> - **Q1: Can an agent output more than one augmentation for a patch?**
>
>     **A1:** The hyperparameter time horizon *T* determines the times of augmentations sequentially performed on a patch.
> Therefore the agent can output more than one augmentation for a patch by setting *T > 1*.
>
> &nbsp;
> - **Q2: "Given the more "local" nature of this augmentation method, have you considered evaluating it on more "local" tasks, such as object detection and semantic segmentation. Perhaps it could make a bigger impact there."**
>
>     **A2:** Thank you for the suggestions. We conduct the experiments on the object detection dataset Pascal VOC 2007. We adopt the mainstream method Faster R-CNN [3] (ResNet-50) as the detector. From the table below we can observe that the proposed PAA is indeed effective for the "local" tasks such as object detection. We have added the results in our revision. In addition, we will consider including more discussion on the "local" nature of PAA.
>
>     |        Method         | mAP (%)  |
>     | :-------------------: | :------: |
>     |     Faster R-CNN      |   76.0   |
>     | Faster R-CNN + Cutout |   77.2   |
>     | Faster R-CNN + Mixup  |   78.3   |
>     |   Faster R-CNN + AA   |   78.7   |
>     |  Faster R-CNN + PAA   | **79.0** |
>
> &nbsp;
> - **Q3: Does each agent's policy is identical and the only difference between the agents is the observation?**
>
>     **A3:** Each agent's policy is not identical, though they share parameters by using a fully convolutional network. When determining an action, each agent's policy relies on different activations of the policy network. A similar design can be found in [1, 2].
>
> &nbsp;
> - **Q4: Do the times from Table 3 include the inference time required to compute the agent state and observations with the auxiliary ResNet-18?**
>
>     **A4:** The training time from Table 3 includes the inference time required to compute the agent state and observations. We will clarify this in the revision.
>
> ---
>
> - **Reply to the suggestions (Per-section):**
>
>     Thank you for your detailed and insightful suggestions.
>
>     **- Related Work:** Thank you for pointing out some missed related works. We have added and discussed them in Related Work in the revision.
>
>     **- Method:** Sorry for the typos and inappropriate words. We have modified them in the revision based on your suggestions.
>
>     **- Experiments:** We adopt your suggestions and revise the "main reasons to reduce" to "main methods for reducing the computational cost" in Section 4.3. In addition, we have further explained Table 3 in our revision, to help readers understand.
>
>     **- Conclusion:** Thank you for the positive comments and useful suggestions. The patch mechanism in our method is naturally aligned with that in vision transformers, and we believe that it will promote the community.
>
> &nbsp;
>
> **Reference**
>
> [1] Chu T, Qu S, Wang J. Large-scale multi-agent reinforcement learning using image-based state representation[C]//2016 IEEE 55th Conference on Decision and Control (CDC). IEEE, 2016: 7592-7597.
>
> [2] Furuta R, Inoue N, Yamasaki T. Pixelrl: fully convolutional network with reinforcement learning for image processing[J]. IEEE Transactions on Multimedia, 2019, 22(7): 1704-1719.
>
> [3] Ren S, He K, Girshick R, et al. Faster r-cnn: Towards real-time object detection with region proposal networks[J]. Advances in neural information processing systems, 2015, 28: 91-99.

---

> ### Author Response · Authors · 2021-11-22
> **Further discussion**
>
> Dear Reviewer XdSw,
>
> Thanks for your detailed and valuable comments! We have updated our revision according to your suggestions.
>
> We would like to send you a friendly reminder that the rolling discussion is coming to an end, and we look forward to knowing whether you have any other questions or comments.
>
> Thanks,
>
> Authors

---

### Official Review · Reviewer_CpsG · 2021-11-02

**Correctness:** 3
**Technical Novelty And Significance:** 2
**Empirical Novelty And Significance:** 2
**Recommendation:** 5
**Confidence:** 4

**Main Review:**

Over all, this paper is well written and technically sound, the experimental evaluation is comprehensive and the results are compelling, but I have several concerns:
1. The most important contribution of this paper in my mind is the new search space focus on image patches for data augmentation. It seems like an incremental job relative to mixup series or cutout series.
2. The proposed PPA formulate the task as a multi-agent reinforcement learning problem, and the results show it can find better policy than random, but [1] demonstrate that only use reinforcement learning can not get better policy than random baseline. The two experimental results are contrary.
3.The calimed SOTA results is inappropriate, methods with better experimental results were not compared. For example, the paper  compare with AdvAA on the  computational cost, but not coapre on the accuracy.
4.Another question is that recent work [2] with careful designed augmentation policy and advanced training strategy can get much better results, It is better to compare with it to give more insights.

[1] Cubuk E D, Zoph B, Shlens J, et al. Randaugment: Practical automated data augmentation with a reduced search space
[2] Wightman R, Touvron H, Jégou H. ResNet strikes back: An improved training procedure in timm

**Summary Of The Paper:**

This paper proposed a fine-grained automated data augmentation approach, Patch AutoAugment (PAA), which tries to increase diversity in local regions by divide an image into a grid of patches and search for the joint optimal augmentation policies for the patches. The proposed PAA considers the task as a multi-agent reinforcement learning problem,  and adopt a multi-agent reinforcement learning
algorithm to automatically search for the optimal augmentation policies by considering the contextual relationship between the patches. They verify the proposed method on many classification and fine-grained recognition dataset(CIFAR10, CIFAR-100, ImageNet, CUB-200-2011, Stanford Cars and FGVC-Aircraft). The experiments show a good result and visualization results provide some insights that the PAA  help the target network to localize more class-related cues.


**Summary Of The Review:**

According to above analysis and concerns, I suggest borderline. If the author can solve my concerns, I will consider change the score.

---

> ### Author Response · Authors · 2021-11-15
> **Response to Reviewer CpsG**
>
> Thank you for your constructive feedback. We would like to address your concerns below.
> - **Q1: "The most important contribution of this paper in my mind is the new search space focus on image patches for data augmentation. It seems like an incremental job relative to mixup series or cutout series."**
>
>     **A1:** Our method is conceptually orthogonal to the works you mentioned, as we discussed in Section 2.1. Our method automates the search of the augmentation policies that choose optimal transformations for images considering the content of patches together with the training status of the target network through using reinforcement learning. However, the mixup and cutout series are manually designed methods that require additional human prior knowledge, and sometimes they are limited to certain datasets. In addition, our method can explore more diversity of DA thanks to our fine-grained transformations, while the mixup and cutout series generally contain only one transformation in an image.
>
>  &nbsp;
> - **Q2: [1] demonstrates that only using reinforcement learning can not get a better policy than random baseline.**
>
>     **A2:** RA [1] is proposed to remove the obstacle in the RL-based methods of using separate proxy tasks (*e.g.,* AutoAugment) and is shown to outperform these methods. However, our method is performed in an online manner, being free of proxy tasks.
>
> &nbsp;
> - **Q3: Compare with AdvAA on accuracy.**
>
>     **A3:** The accuracy results of AdvAA are not reported due to the unavailability of the official source. We re-implement AdvAA and show the comparison results in the table below. Our method consistently outperforms AdvAA which searches DA policies at the image level. We have added the results in the revision.
>
>     |  Dataset  |        Model     | RA[1] | AdvAA[3] |   PAA    |
>     | :-------: | :----------: | :---: | :------: | :------: |
>     | CIFAR-10  |  WRN-28-10   | 97.3  |   97.4   | **97.5** |
>     | CIFAR-10  | SS(26 2x96d) | 97.8  |   97.9   | **98.1** |
>     | CIFAR-100 |  WRN-28-10   | 83.3  |   83.3   | **83.4** |
>     | CIFAR-100 | SS(26 2x96d) | 85.5  |   85.7   | **85.9** |
>     | ImageNet  |  ResNet-50   | 77.6  |   77.9   | **78.3** |
>     |    CUB    |  ResNet-50   | 86.9  |   87.1   | **87.5** |
>     |    CUB    |  ResNet-101  | 88.0  |   88.1   | **88.3** |
>     | Aircraft  |  ResNet-101  | 92.6  |   93.1   | **93.5** |
>
> &nbsp;
> - **Q4: It is better to compare with the carefully designed augmentation methods and advanced training strategy mentioned in [2].**
>
>     **A4:** Thanks for your suggestions. [2] proposes three training procedures with different hyperparameters, denoted as A1, A2 and A3 respectively. We compare the top-1 accuracy on ImageNet with ResNet-50 over Baseline, Mixup, Cutmix, AA, RA [1] and our PAA. As shown in the table below, PAA consistently outperforms other compared methods under different training procedures introduced by [2].
>
>     | Procedures | Baseline | Mixup | CutMix |  AA  | RA[1] |   PAA    |
>     | :--------: | :------: | :---: | :----: | :--: | :---: | :------: |
>     |     A1     |   79.4   | 79.9  |  79.8  | 80.1 | 80.0  | **80.3** |
>     |     A2     |   78.5   | 79.0  |  79.1  | 79.5 | 79.4  | **79.7** |
>     |     A3     |   77.3   | 77.7  |  77.8  | 77.9 | 77.9  | **78.2** |
> &nbsp;
> **Reference**
>
> [1] Cubuk E D, Zoph B, Shlens J, et al. Randaugment: Practical automated data augmentation with a reduced search space[C]//Proceedings of the IEEE/CVF Conference on Computer Vision and Pattern Recognition Workshops. 2020.
>
> [2] Wightman R, Touvron H, Jégou H. ResNet strikes back: An improved training procedure in timm[J]. arXiv preprint arXiv:2110.00476, 2021.
>
> [3] Zhang X, Wang Q, Zhang J, et al. Adversarial autoaugment[J]. arXiv preprint arXiv:1912.11188, 2019.

---

> ### Author Response · Authors · 2021-11-22
> **Further discussion**
>
> Dear Reviewer CpsG,
>
> Thanks for your time in reviewing our paper!
>
> We would like to send you a friendly reminder that the rolling discussion is coming to an end. We really hope our response has addressed your concerns. If you have any other questions or suggestions, we are willing to discuss them further.
>
> Thanks,
>
> Authors

---

> ### Author Response · Authors · 2021-12-03
> **Kindly reminder**
>
> Dear Reviewer CpsG,
>
> We have not received any reply from you. Considering that Reviewer ci48 just commented *"I wrongly thought I already replied to your response"*, we would like to friendly remind you of the discussion in the review period. If you have any concerns or suggestions, please feel free to leave the comments.
>
> Given that the other three reviewers' recommendations for acceptance, we sincerely hope that our work could be recognized by you.
>
> Thanks,
>
> Authors

---

### Official Review · Reviewer_ci48 · 2021-11-02

**Correctness:** 4
**Technical Novelty And Significance:** 3
**Empirical Novelty And Significance:** 3
**Recommendation:** 6
**Confidence:** 3

**Main Review:**

Strengths
+ It seems effective to utilize the MARL algorithm to reduce the computational complexity for the multiple patches.
+ As a result, the computation cost of the proposed algorithm becomes much reduced when it is compared to the previous auto-augmentation algorithms. Furthermore, the algorithm can select the optimal combinations at every epoch without any additional computations, which can improve its generality across the various tasks.

Weaknesses & Questions
- About state and observation. In the proposed algorithm, the state and the observation are obtained by the deep feature from the trained model. My question is the image to obtain the deep feature. When the image is the original image before the determined augmentation, the state and the observation are not affected by the selected action. Then, the best action can be determined just by one simple simulation, and no reinforcement learning algorithm is necessary to find the optimal augmentation. If the augmented images are fed into the model again, the computation cost cannot be reduced as shown in the experiments. What is the detailed sequence to obtain the deep feature?
- Since the deep feature is utilized, its invariant property can affect the determination of the optimal augmentation. In other words, when the trained model is robust to the translation shift, the deep features from the model would be similar even after various translation shifts. Then, the training loss becomes also similar, which results in no selection of translation augmentation. This means that the algorithm would select only the augmentation method that is not considered in the initial model, and this can work as one of the biased optimizations for the optimal augmentation methods. Is there any analysis about the robustness of the initial model for the deep features?
- In the experiments, the proposed algorithm is only compared with the out-of-date algorithms, so its performance cannot be validated well. I recommend the authors add the state-of-the-art studies related to auto-augmentation and the learning-based augmentation algorithms, which include Saliency-mix[1] and Co-mixup[2].
- What would be the limitation of the proposed algorithm?

[1] Uddin et al. "SaliencyMix: A Saliency Guided Data Augmentation Strategy for Better Regularization", ICLR2021
[2] Kim et al. "Co-Mixup: Saliency Guided Joint Mixup with Supermod-ular  Diversity", ICLR2021


**Summary Of The Paper:**

This paper target the task of automatically determining the best augmentation method to obtain improved accuracy. While the previous related studies focus on the image-level augmentation and ignore the semantic information of the augmented images, the proposed algorithm augments the grid-wise patches of the given input with the preserved semantic information. To overcome the enlarged number of combinations to consider all the patches, the algorithm utilizes the MARL algorithm with the unified reward function. By MARL algorithm, the number of parameters can be reduced and the training speed can be much improved, compared to the previous auto-augmentation methods. Through the image classification and fine-grained image recognition tasks, the proposed algorithm was validated, and it shows the state-of-the-art performance among the compared methods.

**Summary Of The Review:**

As I said above, I have some questions about the necessity of the reinforcement learning-based algorithm in the framework. In addition, the comparison with the state-of-the-art algorithms should be added for the validation. I will change my score when I solve the above issues.

---

> ### Author Response · Authors · 2021-11-15
> **Response to Reviewer ci48 (Part 1)**
>
> Thank you for your valuable comments. We address your concerns below.
>
> - **Q1: "About state and observation. In the proposed algorithm, the state and the observation are obtained by the deep feature from the trained model. My question is the image to obtain the deep feature. When the image is the original image before the determined augmentation, the state and the observation are not affected by the selected action. Then, the best action can be determined just by one simple simulation, and no reinforcement learning algorithm is necessary to find the optimal augmentation. If the augmented images are fed into the model again, the computation cost cannot be reduced as shown in the experiments. What is the detailed sequence to obtain the deep feature?"**
>
>     **A1:**      - **Reasons for adopting RL algorithm.** We adopt the RL algorithm for two reasons.
>
>   (i) Learning the optimal DA policies for patches needs to interact with the target network. Given that the nature of RL is to learn to make decisions through interacting with the environment [1, 2], we believe that using RL algorithms is a reliable way of searching policies for patches. Even in the single-step decision-making process, RL can still effectively search for the optimal policy as illustrated in [3, 4].
>
>   (ii) If we don't use RL, the feedback from the target network is difficult to propagate to the augmentation policy network. This is because the augmentation decision-making process (i.e., choosing augmentation operations) is not differentiable.
>
>     **- Detailed sequence to obtain the deep features.** In PAA, there is a hyperparameter time horizon *T* which determines the times of augmentations sequentially performed on a patch.
>  We set the time horizon *T* to 1 for saving computational cost and avoiding over-regularization. We feed the raw image to a pre-trained feature extractor to obtain the deep features as the state/observation, and we then input features to the policy network to get the augmentation operation map. The patches are augmented according to the map and the augmented image is fed into the target network for training. Then, we utilize the status (e.g., the training loss) of the target network as a reward signal to update the policy network.
> &nbsp;
> -  **Q2: "Since the deep feature is utilized, its invariant property can affect the determination of the optimal augmentation. In other words, when the trained model is robust to the translation shift, the deep features from the model would be similar even after various translation shifts. Then, the training loss becomes also similar, which results in no selection of translation augmentation. This means that the algorithm would select only the augmentation method that is not considered in the initial model, and this can work as one of the biased optimizations for the optimal augmentation methods. Is there any analysis about the robustness of the initial model for the deep features?"**
>
>     **A2:** - The pre-trained feature extractor (extracting deep features as the state/observation) and the target network (providing feedback as a reward signal) are two independent parts. We take the mentioned "translation" as an example. We put the original image and the augmented image into the target network and their training losses are not similar (except that the target network has obtained the translation invariance through training), which does not affect the choice of the "translation" operation.
>
>     **- Robustness of the non-pretrained model for the deep features**. We further study the impact of the initialization of the feature extractor as below. We first observe that the performance of the *Non-pretrained ResNet-18* scheme (using non-pretrained ResNet-18 as the state/observation extractor) is still much better than the baseline. We also notice that the performance of the *Non-pretrained ResNet-18* scheme is slightly lower than the *Pretrained ResNet-18*. This may be because that at the very early stage of the training, the features extracted by *Non-pretrained ResNet-18* are not as good as *Pretrained ResNet-18* in practice.
>
>     |    Dataset (Model)    | Baseline | Pretrained ResNet-18 | Non-pretrained ResNet-18 |
>     | :-------------------: | :------: | :------------------: | :----------------------: |
>     | CIFAR-100 (WRN-28-10) |   81.2   |         83.4         |           83.2           |
> &nbsp;
>
> **Reference**
>
> [1] Sutton R S, Barto A G. Reinforcement learning: An introduction[M]. MIT press, 2018.
>
> [2] Arulkumaran K, Deisenroth M P, Brundage M, et al. Deep reinforcement learning: A brief survey[J]. IEEE Signal Processing Magazine, 2017.
>
> [3] Mnih V, Badia A P, Mirza M, et al. Asynchronous methods for deep reinforcement learning[C]// PMLR, 2016.
>
> [4] Singh S, Jaakkola T, Littman M L, et al. Convergence results for single-step on-policy reinforcement-learning algorithms[J]. Machine learning, 2000.

---

> > ### Author Response · Authors · 2021-11-15
> > **Response to Reviewer ci48 (Part 2)**
> >
> > - **Q3: The comparison with more state-of-the-art methods is recommended to include.**
> >
> >     **A3:** Thank you for your suggestions. We compare more state-of-the-art DA methods including Saliency-Mix [1], Co-Mixup [2], RandAugment [3] and DADA [4]. The detailed Top-1 accuracy results are as follows. We have added these comparison results in our revision.
> >
> >     |  Dataset  |    Model     | Saliency-Mix[1] | Co-Mixup[2] | RA[3] | DADA[4] |   PAA    |
> >     | :-------: | :----------: | :-------------: | :---------: | :---: | :-----: | :------: |
> >     | CIFAR-10  |  WRN-28-10   |      97.2       |    97.3     | 97.3  |  97.3   | **97.5** |
> >     | CIFAR-10  | SS(26 2x96d) |      97.8       |    98.0     | 97.8  |  98.0   | **98.1** |
> >     | CIFAR-100 |  WRN-28-10   |      83.3       |  **83.4**   | 83.3  |  82.5   | **83.4** |
> >     | CIFAR-100 | SS(26 2x96d) |      85.6       |    85.8     | 85.5  |  85.7   | **85.9** |
> >     | ImageNet  |  ResNet-50   |      77.9       |    77.6     | 77.6  |  77.5   | **78.3** |
> >     |    CUB    |  ResNet-50   |      87.0       |    87.1     | 86.9  |  86.8   | **87.5** |
> >     |    CUB    |  ResNet-101  |      88.1       |    88.2     | 88.0  |  88.1   | **88.3** |
> >     | Aircraft  |  ResNet-101  |      92.8       |    93.1     | 92.6  |  92.8   | **93.5** |
> > &nbsp;
> > - **Q4: What would be the limitation of the proposed algorithm?**
> >
> >     **A4:** Compared with image-level DA methods, our method has an extra hyperparameter N (number of divided patches) to adjust.
> >
> > &nbsp;
> >
> > **Reference**
> >
> > [1] Uddin A F M, Monira M, Shin W, et al. Saliencymix: A saliency guided data augmentation strategy for better regularization[J]. arXiv preprint arXiv:2006.01791, 2020.
> >
> > [2] Kim J H, Choo W, Jeong H, et al. Co-mixup: Saliency guided joint mixup with supermodular diversity[J]. arXiv preprint arXiv:2102.03065, 2021.
> >
> > [3] Cubuk E D, Zoph B, Shlens J, et al. Randaugment: Practical automated data augmentation with a reduced search space[C]//Proceedings of the IEEE/CVF Conference on Computer Vision and Pattern Recognition Workshops. 2020.
> >
> > [4] Li Y, Hu G, Wang Y, et al. DADA: Differentiable automatic data augmentation[J]. arXiv preprint arXiv:2003.03780, 2020.

---

> ### Author Response · Authors · 2021-11-22
> **Further discussion**
>
> Dear Reviewer ci48,
>
> We sincerely thank you for your efforts in reviewing our paper and the valuable suggestions for improving our paper!
>
> We would like to send you a friendly reminder that the rolling discussion is coming to an end, and we look forward to knowing whether our response has addressed your concerns. If there is any other questions or comments, please feel free to discuss them with us.
>
> Thanks,
>
> Authors

---

### Author Response · Authors · 2021-11-17
**Summary of changes in revision**

We thank all the reviewers for their helpful comments and feedback. We have updated our work based on the suggestions.

Major updates are summarized below:
- Some missed related works are added and discussed in Section 2.1.
- The experimental comparison results of more SOTA DA methods are provided in Table 1,2,4.
- Some details and explanations of the experimental setting are added in Appendix C and D.
- More empirical studies on the initialization of the feature extractor and effectiveness of PAA on local tasks (e.g., object detection) in Appendix F and G.

All the modifications are marked with red. We will address individual concerns and questions in the responses.
Further updates made in the following revisions will be informed.

Thanks,

Authors

---

### Author Response · Authors · 2021-11-20
**Please let us know if our response has addressed your concerns**

We thank all the reviewers for their great efforts in reviewing our paper! We are looking forward to knowing whether our response has well addressed your concerns.

Please feel free to leave your comments if you have any further questions or suggestions.

---

### Decision · Program_Chairs · 2022-01-20

**Decision:**

Reject

**Comment:**

The paper proposed an approach to search image augmentation policies. The paper formulates this problem as a cooperative multi-agent decision-making problem, which is interesting. The paper received 3 borderline accept and 1 borderline reject ratings. The reviewers originally had multiple concerns regarding the necessity of RL-based approach, lacking references, and additional experiments, and the authors responded to some of the concerns of the reviewers reasonably. However, none of the reviewers ended up strongly supporting the paper, staying with their ratings.

The RL formulation of the problem is interesting, but it requires multiple rounds of the target network training due to its nature (i.e., it is not an end-to-end approach). The paper misses some details on how exactly the patch-wise RL-based augmentation works and it requires additional hyperparameters for the selection of patch size and shape. It is also unclear how this RL-based method is conceptually superior to previous augmentation approaches and the empirical results are not strong enough, as some of the reviewers also pointed out.

Although the paper has interesting ideas and the AC also think the paper has some merit, the senior AC finds the technical contribution of the paper weaker than the others. We unfortunately need to recommend the rejection of the paper.